# Sequence variants in *ARHGAP15*, *COLQ* and *FAM155A* associate with diverticular disease and diverticulitis

Snaevar Sigurdsson[1], Kristjan F. Alexandersson[1], Patrick Sulem[1], Bjarke Feenstra[2], Steinunn Gudmundsdottir[1], Gisli H. Halldorsson[1], Sigurgeir Olafsson[1], Asgeir Sigurdsson[1], Thorunn Rafnar[1], Thorgeir Thorgeirsson[1], Erik Sørensen[3], Andreas Nordholm-Carstensen[4], Jakob Burcharth[5], Jens Andersen[6], Henrik Stig Jørgensen[7], Emma Possfelt-Møller[8], Henrik Ullum[3], Gudmar Thorleifsson[1], Gisli Masson[1], Unnur Thorsteinsdottir[1,9], Mads Melbye[2,10,11], Daniel F. Gudbjartsson[1,12], Tryggvi Stefansson[13], Ingileif Jonsdottir[1,9,14] & Kari Stefansson[1,9]

Diverticular disease is characterized by pouches (that is, diverticulae) due to weakness in the bowel wall, which can become infected and inflamed causing diverticulitis, with potentially severe complications. Here, we test 32.4 million sequence variants identified through whole-genome sequencing (WGS) of 15,220 Icelanders for association with diverticular disease (5,426 cases) and its more severe form diverticulitis (2,764 cases). Subsequently, 16 sequence variants are followed up in a diverticular disease sample from Denmark (5,970 cases, 3,020 controls). In the combined Icelandic and Danish data sets we observe significant association of intronic variants in *ARHGAP15* (Rho GTPase-activating protein 15; rs4662344-T: $P = 1.9 \times 10^{-18}$, odds ratio (OR) = 1.23) and *COLQ* (collagen-like tail subunit of asymmetric acetylcholinesterase; rs7609897-T: $P = 1.5 \times 10^{-10}$, OR = 0.87) with diverticular disease and in *FAM155A* (family with sequence similarity 155A; rs67153654-A: $P = 3.0 \times 10^{-11}$, OR = 0.82) with diverticulitis. These are the first loci shown to associate with diverticular disease in a genome-wide study.

[1] deCODE Genetics/Amgen Inc., Sturlugata 8, 101 Reykjavik, Iceland. [2] Department of Epidemiology Research, Statens Serum Institut, 2300 Copenhagen, Denmark. [3] Department of Clinical Immunology, Copenhagen University Hospital/Rigshospitalet, 2100 Copenhagen, Denmark. [4] Digestive Disease Center, Bispebjerg Hospital, University of Copenhagen, Copenhagen, 2400 Nevada, Denmark. [5] Department of Surgery, Herlev Hospital, University of Copenhagen, 2730 Herlev, Denmark. [6] Hvidovre university Hospital, Department of Surgery. Gastroenterology, 2650 Hvidovre, Denmark. [7] Department of Surgery, Northern Sealand Hospital, 3400 Hillerød, Denmark. [8] Department of Surgical Gastroenterology, Copenhagen University Hospital/Rigshospitalet, 2100 Copenhagen, Denmark. [9] Faculty of Medicine, School of Health Sciences, University of Iceland, 101 Reykjavik, Iceland. [10] Department of Clinical Medicine, University of Copenhagen, 2100 Copenhagen, Denmark. [11] Department of Medicine, Stanford University School of Medicine, Stanford, California 94305-5475, USA. [12] School of Engineering and Natural Sciences, University of Iceland, 101 Reykjavik, Iceland. [13] Department of Surgery, Landspitali, the National University Hospital of Iceland, 101 Reykjavik, Iceland. [14] Department of Immunology, Landspitali, the National University Hospital of Iceland, 101 Reykjavik, Iceland. Correspondence and requests for materials should be addressed to I.J. (email: ingileif.jonsdottir@decode.is) or to K.S. (email: kstefans@decode.is).

Diverticular disease is thought to be due to complex interactions between diet, lifestyle, colonic motility, structural changes in the gut, enteric neuropathy and smoking[1–3]. The intestinal microflora and low-grade inflammation may contribute to diverticular disease and acute diverticulitis[3]. Diverticulae are commonly found during routine colonoscopy with increased prevalence from age 50–59 years (32.6%) to age ≥80 years (71.4%)[4,5]. Up to 20% will experience complications of the disease[6] but only 1–4% of individuals with diverticula develop acute diverticulitis, with a recurrence risk of 20% within 5 years[7].

Relative risk of siblings of diverticular disease cases is 2.9 (ref. 8) and the heritability in twin studies estimated to be 40–50% (refs 8,9). This indicates that there is a strong genetic component to the risk. No sequence variants associating with risk of diverticular disease have been found and no genome-wide association studies (GWAS) have been published.

We performed GWAS to search for sequence variants that affect the risk of diverticular disease and diverticulitis in Iceland with a follow up in Danish sample of diverticular disease. We find association of three intronic variants in the genes *ARHGAP15* (Rho GTPase-activating protein 15) and *COLQ* (collagen-like tail subunit of asymmetric acetylcholinesterase) with diverticular disease and in *FAM155A* (family with sequence similarity 155A) with diverticulitis. These are the first sequence variants found to show genome-wide significant association with diverticular disease.

## Results

**Association of three loci with diverticular disease**. We imputed 32.4 million sequence variants identified through WGS of 15,220 Icelanders into 151,677 chip typed Icelanders and their first- and second-degree relatives[10,11] and performed two GWAS to search for sequence variants that affect the risk of diverticular disease (5,426 cases) and diverticulitis (2,764 cases) using the same 245,951 controls (Supplementary Table 1). We applied weighted thresholds for genome-wide significance that depend on the functional class of each variant, based on its prior probability of affecting gene function[12] (Supplementary Table 2).

We chose 16 variants for follow-up in a Danish diverticular disease sample set that were within two orders of magnitude from genome-wide significance threshold in a variant class for either diverticular disease or diverticulitis in Iceland (Table 1 and Supplementary Table 3a–c). We do not have information on diverticulitis in the Danish cohort, but chose to follow-up the Icelandic diverticulitis findings based on the assumption that the Danish cohort includes diverticulitis, although the proportion is unknown. With these data sets we identified three loci that are of genome wide-significance in the combined analysis of the Icelandic and Danish samples; intronic variants at the *ARHGAP15* and *COLQ* loci associate significantly with diverticular disease and at *FAM155A* locus with diverticulitis ($P < 2.3 \times 10^{-9}$, the threshold for intronic variants within a DNase hypersensitivity site[12]) (Table 1 and Supplementary Table 3). No significant heterogeneity was observed between the study groups and the three singe-nucleotide polymorphisms (SNPs) are nominally significant in the Danish follow-up.

**Potential causal genes at the diverticular disease loci**. The strongest diverticular disease association in Iceland was with 45 correlated ($r^2 > 0.97$) sequence variants (minor allele frequency (MAF) = 17.6–17.8%) in 88 kb region spanning introns 9 and 10 of *ARHGAP15* (Rho GTPase-activating protein 15) (Fig. 1a). The variants are represented by rs4662344-T that associates at genome-wide significance in Iceland (chr2:143,591,289,

odds ratio (OR) = 1.23, $P = 4.9 \times 10^{-13}$) (Table 1). In Iceland rs4662344-T confers similar risk of diverticulitis (OR = 1.26, $P = 4.5 \times 10^{-9}$) and uncomplicated diverticular disease (OR = 1.20, $P = 2.6 \times 10^{-6}$) ($P_{het} = 0.36$). No association signal remains at the locus after conditional analysis using rs4662344-T as a covariate (Fig. 2a), indicating that one of the 45 intronic variants is likely to mediate the signal at this locus. The association replicates in the Danish samples (OR = 1.22 and $P = 7.3 \times 10^{-7}$) for a combined $P$ value of $1.9 \times 10^{-18}$ and OR of 1.23 for the Icelandic and Danish samples. None of the three missense variants in *ARHGAP15* (15 exons 475 amino acids) associate with diverticular disease ($P > 0.44$). No other gene is within 100 kb of any of the SNPs in linkage disequilibrium (LD) ($R^2 > 0.8$) with rs4662344-T.

rs4662344-T did not associate with expression of *ARHGAP15* or any other gene in the region (± 500 kb) in any of the tissues in GTExV6 database (including whole blood and small intestine, the most relevant tissues) nor in RNA sequencing data at deCODE from whole blood ($n = 2,246$) and adipose tissue ($n = 708$) (Supplementary Fig. 1, shown for each transcript and each exon).

*ARHGAP15* encodes Rho GTPase-activating protein 15, a member of the Rac-specific GTPase-activating protein (GAP). Rac is a small GTPase, important for cell proliferation, apoptosis, attachment and motility[13]. ARHGAP15s activation of Rac affects the actin cytoskeleton and cell morphogenesis[14] and overexpression of *ARHGAP15* causes increase in actin stress fibres and cell contraction[15]. Neutrophils of mice that are knockout for *ArhGAP15* show increased migration, phagocytosis, reactive oxygen species (ROS) production and bacterial killing, and reduced inflammation[13]. We therefore tested the effect of rs4662344-T on ROS production by neutrophils stimulated with *E. coli*, phorbol 12-myristate 13-acetate or *N*-formyl-MetLeuPhe, but found no effect of rs4662344-T carrier status on neutrophil ROS production for any of the stimulants (Supplementary Fig. 2).

The second strongest signal that associates with diverticular disease in the Icelandic samples is captured by a single intronic SNP in *COLQ*, rs7609897-T (chr3:15,461,174, MAF = 24.7%) (Fig. 2b), that associates with diverticular disease (OR = 0.85, $P = 1.6 \times 10^{-9}$) in Iceland; this association replicates nominally in the Danish samples with a consistent direction of the effect (OR = 0.91, $P = 0.010$) for a combined OR of 0.87 and $P$ value of $1.5 \times 10^{-10}$ for the Icelandic and Danish samples. In the Icelandic and in 1,000G European data sets, rs7609897-T is weakly correlated with other markers ($r^2 < 0.26$ and $r^2 < 0.48$, respectively). We performed conditional analysis to look for additional signals at the locus (Supplementary Table 5). We found one rare missense variant rs146687198-G (p.Gly246Ala, MAF = 0.22%) in *COLQ* with large effect on diverticular disease in Iceland (OR = 2.06, 95% confidence interval (CI): 1.4, 3.0, $P = 3.5 \times 10^{-4}$). This rare missense variant is not correlated with the intronic rs7609897-T ($r^2 < 0.001$). Follow-up genotyping in the Danish samples showed a weaker and not significant effect (OR = 1.15, 95% CI: 0.63, 2.10, $P = 0.65$, MAF = 0.26%). However, the effect is consistent in the two populations ($P_{het} = 0.11$, for Iceland and Danish samples). Although the association of this rare missense variant with diverticular disease is not of genome-wide significance, the prior probability established by the association of rs7609897-T suggests that the association of rs146687198-G may be real and points to *COLQ* as the causative gene at this locus.

*COLQ* has 18 exons that span a 71 kb region (Supplementary Table 5). *COLQ* is expressed in most tissues (GTEx V6)[16]. rs7609897-T did not associate with RNA expression of *COLQ* or any of the 11 genes within 500 kb of rs7609897-T in any of the tissues in the GTExV6 database or in the deCODE in blood or adipocyte RNA sequencing data (Supplementary Fig. 3).

**Table 1 | Icelandic GWAS results, follow-up in a Danish diverticular disease sample set and association in the Icelandic and Danish sample sets combined.**

| Nearest gene | SNP | Amin/Amaj | Icelandic diverticular disease | | | Danish diverticular disease* | | | Combined Icelandic and Danish diverticular disease sample sets | | |
| --- | --- | --- | --- | --- | --- | --- | --- | --- | --- | --- | --- |
| | | | 5,426 cases; 245,951 controls | | | 5,970 cases; 3,020 controls | | | 11,396 cases; 248,971 controls | | |
| | | | MAF % cases–controls | OR (95% CI) | P value | MAF % cases–controls | OR (95% CI) | P value | OR (95% CI) | P value | $P_{het}$ |
| ARHGAP15 | rs4662344 | T/C | 20.9/17.7 | 1.23 (1.16, 1.31) | $4.9 \times 10^{-13}$ | 21.7/18.5 | 1.22 (1.13, 1.32) | $7.0 \times 10^{-7}$ | 1.23 (1.17, 1.29) | $1.9 \times 10^{-18}$ | 0.86 |
| COLQ | rs7609897 | T/G | 22.1/24.7 | 0.85 (0.80, 0.89) | $1.6 \times 10^{-9}$ | 21.6/23.3 | 0.91 (0.84, 0.98) | $1.0 \times 10^{-2}$ | 0.87 (0.83, 0.91) | $1.5 \times 10^{-10}$ | 0.17 |
| FAM155A | rs67153654 | A/T | 17.0/18.6 | 0.89 (0.84, 0.94) | $8.7 \times 10^{-5}$ | 17.4/20.1 | 0.84 (0.78, 0.91) | $2.2 \times 10^{-5}$ | 0.87 (0.83, 0.91) | $1.3 \times 10^{-8}$ | 0.30 |

| Nearest gene | SNP | Amin/Amaj | Icelandic diverticulitis | | | Danish diverticular disease* | | | Combined Icelandic diverticulitis and Danish diverticular disease sample sets | | |
| --- | --- | --- | --- | --- | --- | --- | --- | --- | --- | --- | --- |
| | | | 2,764 cases; 245,951 controls | | | 5,970 cases; 3,020 controls | | | 8,734 cases; 248,971 controls | | |
| | | | MAF % cases–controls | OR (95% CI) | P value | MAF % cases–controls | OR (95% CI) | P value | OR (95% CI) | P value | $P_{het}$ |
| ARHGAP15 | rs4662344 | T/C | 21.3/17.7 | 1.26 (1.16, 1.36) | $4.5 \times 10^{-9}$ | 21.7/18.5 | 1.22 (1.13, 1.32) | $7.0 \times 10^{-7}$ | 1.24 (1.17, 1.31) | $1.8 \times 10^{-14}$ | 0.62 |
| COLQ | rs7609897 | T/G | 21.0/24.7 | 0.8 (0.74, 0.86) | $1.9 \times 10^{-9}$ | 21.6/23.3 | 0.91 (0.84, 0.98) | $1.0 \times 10^{-2}$ | 0.85 (0.80, 0.89) | $1.0 \times 10^{-9}$ | 0.02 |
| FAM155A | rs67153654 | A/T | 15.6/18.6 | 0.8 (0.74, 0.87) | $2.3 \times 10^{-7}$ | 17.4/20.1 | 0.84 (0.78, 0.91) | $2.2 \times 10^{-5}$ | 0.82 (0.78, 0.87) | $3.0 \times 10^{-11}$ | 0.43 |

| Nearest gene | SNP | Amin/Amaj | Icelandic uncomplicated diverticular disease | | | Danish diverticular disease* | | | Combined uncomplicated diverticular disease and Danish diverticular disease sample sets | | |
| --- | --- | --- | --- | --- | --- | --- | --- | --- | --- | --- | --- |
| | | | 2,662 cases; 245,951 controls | | | 5,970 cases; 3,020 controls | | | 8,632 cases; 248,971 controls | | |
| | | | MAF % cases/controls | OR (95% CI) | P value | MAF % cases/controls | OR (95% CI) | P value | OR (95% CI) | P value | $P_{het}$ |
| ARHGAP15 | rs4662344 | T/C | 20.4/17.7 | 1.2 (1.11, 1.30) | $2.6 \times 10^{-6}$ | 21.7/18.5 | 1.22 (1.13, 1.32) | $7.0 \times 10^{-7}$ | 1.21 (1.15, 1.28) | $8.6 \times 10^{-12}$ | 0.77 |
| COLQ | rs7609897 | T/G | 23.7/24.7 | 0.91 (0.84, 0.97) | $6.1 \times 10^{-3}$ | 21.6/23.3 | 0.91 (0.84, 0.98) | 0.01 | 0.91 (0.86, 0.95) | $1.7 \times 10^{-4}$ | 1.0 |
| FAM155A | rs67153654 | A/T | 18.87/18.6 | 0.99 (0.91, 1.06) | 0.72 | 17.4/20.1 | 0.84 (0.78, 0.91) | $2.2 \times 10^{-5}$ | 0.91 (0.87, 0.97) | $1.4 \times 10^{-3}$ | $4.9 \times 10^{-3}$ |

Amaj, major allele; Amin, minor allele; 95% CI, 95% confidence interval; MAF, minor allele frequency; OR, odds ratio of the minor allele; $P_{het}$, P value for the heterogeneity between cohorts; SNP, single-nucleotide variant polymorphism.
MAF is calculated on the chip-typed samples, excluding familial imputed genotypes. The three variants rs4662344, rs7609897 and rs67153654 are annotated as intronic variant within a DNase hypersensitivity site, giving the class-specific Bonferroni threshold for genome-wide significance as $P < 2.3 \times 10^{-9}$ (ref.12).
*Note that the Danish diverticular disease results are the same in all three parts of the table, since diverticulitis diagnosis was not available.

COLQ encodes a subunit of a collagen-like molecule (ColQ) associated with acetylcholinesterase (AChE), whose catalytic subunits are anchored in the basal lamina of neuromuscular junction through ColQ. Homozygote mutations (or compound heterozygotes) in COLQ can reduce AChE availability resulting in prolonged nerve to muscle signalling, that can cause muscle weakness and congenital myasthenic syndromes[17].

The third locus harbours sequence variants in the first intron of FAM155A, marked by rs67153654-A, showing a suggestive association with diverticulitis in Iceland (Fig. 1b and Fig. 2c) (chr13:107,572,636, MAF = 18.6%, OR = 0.80, 95% CI: 0.74, 0.87, $P = 2.3 \times 10^{-7}$). Although diagnosis of diverticulitis is not available for the Danish samples, we genotyped the FAM155A variant in the Danish diverticular disease samples. We replicated the association with rs67153654-A (Danish diverticular disease: OR = 0.84, $P = 2.2 \times 10^{-5}$) despite the lack of information on the proportion of diverticulitis among the Danish diverticular disease samples. In the combined analysis of the Icelandic diverticulitis and Danish total diverticular disease sample set, rs67153654-A reached genome-wide significance ($P = 3.0 \times 10^{-11}$, OR = 0.82) (Table 1). The association of rs67153654-A is driven by those in the diverticular disease sample who have developed diverticulitis, with no association with the subset of uncomplicated diverticular disease in Iceland (OR = 0.99, 95% CI: 0.92, 1.07, $P = 0.78$), suggesting that this variant is not likely to influence the integrity of the wall of the colon, but rather protection from infection or inflammation.

FAM155A spans 703 kb with only three exons and no other protein coding genes lie within 500 kb of rs67153654-A. None of the 16 missense variants found in FAM155A associates with diverticular disease or diverticulitis (Supplementary Table 5). FAM155A is mainly expressed in the hypothalamus and pituitary gland[18,19] with low expression in the colon and blood (GTEx V6)[16]. We found no effect of rs67153654-A on the expression of FAM155A or nearby genes in GTExV6 or in blood or adipocytes using RNA sequencing (Supplementary Fig. 4). Little is known about the function of FAM155A but close SNPs ($r^2 < 0.01$ with rs67153654-A) have been associated with increased fat mass in children[20] and anorexia nervosa[21]. We tested the association of diverticulitis versus uncomplicated diverticular disease for variants at the ARHGAP15, COLQ and FAM155A loci. (Supplementary Table 4). Only the FAM155A variant is significantly less frequent in diverticulitis (OR = 0.84, 95% CI: 0.74, 0.94, $P = 3.8 \times 10^{-3}$).

The 13 variants at the other loci selected for validation in the Danish samples lack evidence for association with either diverticular disease or diverticulitis (Supplementary Table 3a–c and Supplementary Note).

Inflammation contributes to the development and recurrence of diverticulitis[22]. Therefore, we tested the effects of the diverticular disease variants at the ARHGAP15, COLQ and FAM155A loci on other inflammatory diseases of intestine and colon, namely ulcerative colitis (UC) and Crohn's disease (CD) (inflammatory bowel disease (IBD)) and found no association with a $P < 1 \times 10^{-3}$. Furthermore, none of the 184 IBD/UC/CD variants (from the GWAS catalogue)[23–29] associate with diverticulitis or diverticular disease ($0.05/184 = P < 2.7 \times 10^{-4}$) in Iceland. Neither did polygenic risk scores (PRS) for IBD, UC and CD capture risk of diverticular disease or diverticulitis (Supplementary Table 6).

Few sequence variants have previously been reported to associate with diverticular disease, diverticulitis or diverticular disease -related diseases in small candidate gene studies[30–33]. We show no evidence for association of these variants with the disease in the Icelandic data (Supplementary Table 7).

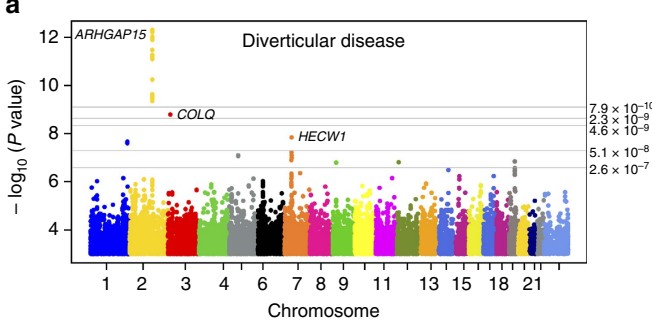

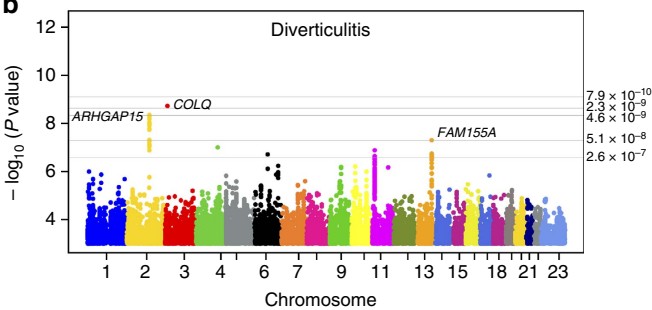

**Figure 1 | Manhattan plot for genome-wide association results.** The P values ($-\log_{10}$) are plotted against their respective positions on each chromosome. P value thresholds for the different annotation classes are indicated with gray lines. For intronic/intergenic variants outside DNAse hypersensitivity site: $P = 1.1 \times 10^{-9}$, intronic/intergenic variants within DNAse hypersensitivity site $P = 2.3 \times 10^{-9}$, low-impact variants: $P = 5.3 \times 10^{-9}$, medium-impact variants: $P = 7.4 \times 10^{-8}$ and high-impact variants: $P = 3.7 \times 10^{-7}$. The plots were created using qqman: an R package for visualizing GWAS results using Q–Q and Manhattan plots[54]. (**a**) GWAS results for diverticular disease (cases $n = 5,292$; controls $n = 245,951$). ARHGAP15 $P = 5.1 \times 10^{-12}$, COLQ $P = 2.8 \times 10^{-9}$ and FAM155A $P = 8.0 \times 10^{-5}$. (**b**) GWAS results for diverticulitis (cases $n = 2,764$ cases; controls $n = 245,951$) (excluding diverticular disease). ARHGAP15 $P = 6.0 \times 10^{-9}$, COLQ $P = 3.0 \times 10^{-9}$ and FAM155A $P = 1.7 \times 10^{-7}$. The three variants rs4662344, rs7609897 and rs67153654 are annotated as intronic variant within a DNase hypersensitivity site, giving the class-specific Bonferroni threshold for genome-wide significance as $P < 2.3 \times 10^{-9}$ (ref. 12).

## Discussion

We have found common sequence variants in introns of the ARHGAP15, COLQ and FAM155A that associate with risk of diverticular disease or diverticulitis. These sequence variants do not overlap with known GWAS signals in other diseases or traits, including established risk loci for immune-mediated and inflammatory diseases[34]. Diverticulitis occurs when the mucosa of diverticula becomes inflamed. Often the flat colon mucosa between the orifices of the diverticula is inflamed, with changes indistinguishable from those of UC or CD[35]. We found no genetic overlap between diverticular disease and UC and CD. We found no association of the diverticular disease variants reported here with these diseases and well-established risk variants, and PRS for IBD, UC and CD do not associate with diverticular disease or diverticulitis. This indicates that the pathogenic mechanisms differ from those of autoimmune diseases of the colon and intestine. This is further supported by the complete lack of association with the HLA region that associates strongly with IBD and UC[23].

The stronger association of the FAM155A variants with diverticulitis than diverticular disease in general may reflect effects on disease progression, such as inflammation or infection. Various inflammatory components have been suggested as biomarkers of diverticular disease and diverticulitis, including C-reactive protein, white blood cell count, erythrocyte sedimentation rate and faecal calprotectin[36]. Still we found that the FAM155A variants have no effect on C-reactive protein levels, white blood cell count or neutrophil count, erythrocyte sedimentation rate ($P < 10^{-3}$) (Supplementary Methods), neither do the ARHGAP15 and COLQ variants. Despite the role of Rho GTPase-activating protein 15, encoded by ARHGAP15, on phagocyte function and inflammation the ARHGAP15 variant associating with diverticular disease does not affect ROS production by neutrophils. Whether the diverticular disease variants mediate their effects by modulating inflammation is thus unclear. None of the diverticular disease variants affect the expression of ARGHAP15, COLQ or FAM155A or of nearby genes, neither in deCODE's RNAseq data on blood and adipocytes nor in data from the various tissues of the GTEx database. Thus, the mechanism by which they affect the risk of diverticular disease remains to be elucidated.

Smoking is a risk factor for symptomatic diverticular disease in both men and women increasing risk of developing complicated diverticular disease[1] and hospital admission for acute colonic diverticulitis[37]. We found that heavy smokers ($N = 26,113$, $> 10$ pack-years)[38,39] have higher risk of developing diverticular disease than never smokers ($N = 22,815$) (Supplementary Methods), with relative risk $= 1.35$; 95% CI: 1.21–1.51, $P = 1.11 \times 10^{-7}$ (adjusted for sex and age). However, smoking showed no interaction with the effect of any of the three diverticular disease variants.

This first genome-wide association scan may pave the way for studies on the mechanism underlying the development of diverticular disease and diverticulitis.

## Methods

**Discovery cohort.** We have collected phenotype data on the Icelandic population from everyone diagnosed with diverticular disease (ICD 9: 562.1 − 2 and ICD 10 K57.2 − 9) at the National University Hospital and from Akureyri Hospital in northern Iceland during the years 1985–2014. Phenotype data was available for 5,777 individuals with diverticular disease, including 2,923 with the primary diagnosis of diverticulitis and 2,854 with uncomplicated diverticular disease. Patients who came to the hospital primary for diverticulitis complications or if the diagnosis was coupled to a resection of the left colon or sigmoid colon were classified as diverticulitis. Genotype information was available for 94% of the diverticular disease cases or 2,764 of the 2,923 with diverticulitis and 2,662 of 2,854 uncomplicated diverticular disease. The diverticular disease patients were 60% female, with the mean age of 67.7 years (s.d. = 4.7) and mean body mass index of 27.8 (s.d. = 5.2). Information on the patients is summarized in Supplementary Table 1.

The study was approved by the National Bioethics Committee (ref. VSN 12-121) and the Data Protection Authority (2013030423ÞS/--) in Iceland. All participating subjects who donated blood signed informed consent. Personal identities of the participant's data and biological samples were encrypted by a third-party system (Identity Protection System), approved and monitored by the Data Protection Authority.

**Follow-up cohort.** Statens Serum Institut (SSI) hosts the Danish National Biobank and one of the associated biobanks under the Danish National Biobank umbrella is the Copenhagen Hospital Biobank, which stores EDTA whole blood from patient samples submitted for blood typing at hospitals in the Capital Region of Denmark (the Greater Copenhagen Area). SSI identified 6,500 individuals with diverticular disease diagnosis (ICD 9: 562.1 − 2 and ICD 10 K57.2 − 9), with EDTA whole-blood samples in the Copenhagen Hospital Biobank and further 3,000 control individuals who have no records of DD. Information on the proportion of diverticulitis was not available The study was approved by the Scientific Ethics Committee of the Capital Region of Denmark (H-15000405) and the Danish Data Protection Agency. The Scientific Ethics Committee granted exemption from obtaining informed consent from participants as the study was based on the biobank material. The DNA extraction from whole blood and marker genotyping was performed at deCODE genetics.

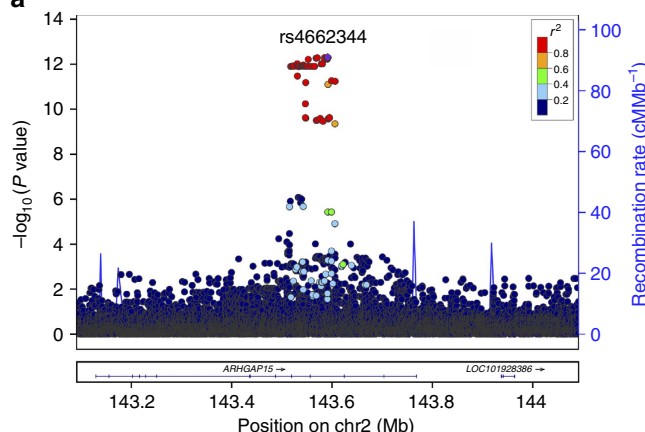

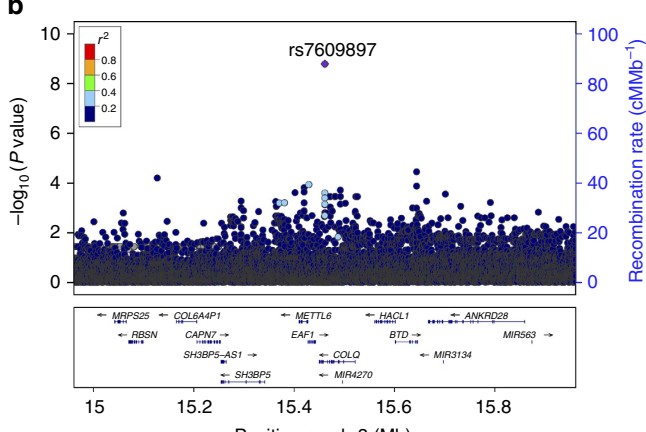

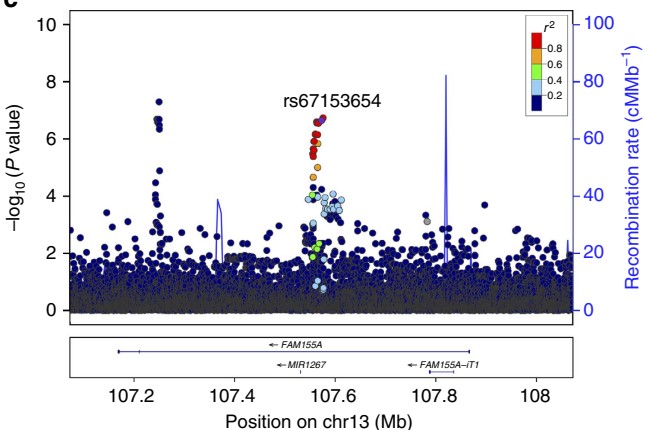

**Figure 2 | Regional association plot for the three associated loci.** P values (−log₁₀) for the marker associations are plotted against the chromosomal location (human genome build 38) at each locus. The colour of the genomic variants reflects the linkage disequilibrium ($r^2$ LD) with the lead SNP in the Icelandic dataset. The blue line indicates recombination rates from the Icelandic recombination map for males and females[55]. Known genes and exons are shown below using data from the UCSC genes track. The plot was created with a stand-alone version of the LocusZoom Software[56]. (**a**) Locus plot for the marker rs4662344-T (chr2:143,591,289) at the *ARHGAP15* locus. P values plotted are for association with diverticular disease in Iceland. (**b**) Locus plot for the marker rs7609897-T (chr3:15,461,174) in intron of the *COLQ* gene. P values plotted are for the association with diverticular disease in Iceland. (**c**) Locus plot for the marker rs67153654-A (chr13:107,572,636) at the *FAM155A* locus. P values plotted are for the association with diverticulitis in Iceland.

**Genotyping and association.** Genotyping and imputation methods and the association analysis in the Icelandic samples was performed as follows: In brief, we sequenced the whole genome of 15,220 Icelanders using Illumina sequencers to a mean depth of at least $\times 10$ (mean 30, median $\times 32$)[11], using three different library preparation methods from Illumina: (a) the standard TruSeq DNA library preparation method; Illumina GAIIx and/or HiSeq 2000 sequencers; (b) the TruSeq DNA PCR-free library preparation method; Illumina HiSeq 2500 sequencers; and (c) the TruSeq Nano DNA library preparation method; Illumina HiSeq X sequencers (see Supplementary Methods for a detailed description of the sequencing methods)[11]. Genotypes of SNPs and indels were called using joint calling with the Genome Analysis Toolkit HaplotypeCaller (GATK version 3.3.0)[40]. Using information about haplotype sharing genotype calls were improved, taking advantage of chip-typing and long-range phasing of all the sequenced individuals. In total, 32,463,443 genetic variants were called ( info > 0.8 and MAF > 0.01%). SNPS and indels that met the quality criteria were imputed into the 151,677 chip-typed Icelanders with the help of extensive genealogical information and long-range phased haplotypes[11]. The sequence variants were imputed into 294,212 untyped relatives of the chip-typed individuals to further increase the sample size for association analysis and increase the power to detect associations[10,11,41].

We used the variant effect predictor[42] to predict the maximal consequence of each sequence variant on all neighbouring RefSeq genes[43]. There is a substantial variation in the enrichment of phenotype-associating sequence variants based on their annotations[12]. On the basis of these enrichments, it is possible to group sequence variants into categories, in order of decreasing impact on biological function. We used the enrichment of variant classes to correct the threshold for genome-wide significance with a weighted Bonferroni adjustment[12]. With 32,463,443 sequence variants tested, the weights given in Sveinbjornsson *et al.*[12] were rescaled to control the family-wise error rate (Supplementary Table 2). This yielded significance thresholds of $P < 2.6 \times 10^{-7}$ for high-impact variants ($N = 8.474$, including stop gained, frameshift, splice acceptor or donor), $P < 5.1 \times 10^{-8}$ for moderate-impact variants ($N = 149,983$, including missense, splice-region variants and in-frame INDELs), $P < 4.6 \times 10^{-9}$ for low-impact variants ($N = 2,283,889$, including synonymous variants 3′- and 5′-untranslated region variants), $P < 2.3 \times 10^{-9}$ for other variants overlapping DNase hypersensitivity sites ($N = 3,913,058$) and $P < 7.9 \times 10^{-10}$ for other non-DNase hypersensitivity site variants, intergenic and deep intronic ($N = 26,108,039$)[12] (Supplementary Table 2). For association testing in the case–control analysis, we used logistic regression; disease status was treated as the response and genotype counts were used as covariates. We also included in the model as nuisance variables the following available individual characteristics that correlate with disease status; county of birth, sex, current age or age of death (first- and second-order terms included), availability of blood sample for the individual and an indicator function for the overlap of the timespan of phenotype collection with lifetime of the individual[11,44,45]. We applied LD score regression to estimate a correction factor to distinguish polygenicity from population stratification in the GWAS results[46]. To correct for the relatedness of the Icelandic individuals included in this study, we applied the method of genomic control[47] where the inflation in the $\chi^2$ values was estimated on the basis of a subset of about 300,000 common variants, and P values were adjusted by dividing the corresponding $\chi^2$ values by this factor. For the diverticular disease, this factor was 1.18 and for diverticulitis 1.17.

A total of 5,426 individuals with diverticular disease were included in the association analysis; 3,368 of these were genotyped using various Illumina chips and imputed using long-range phased haplotypes and the remaining 1,958 were imputed on the basis of genotypes of first- and second-degree relatives[11]. The same population controls, 245,951 individuals recruited through different deCODE projects, were used for association analysis of the three diverticular disease phenotypes: 124,228 genotyped and 121,723 imputed on the basis of genotypes of first- and second-degree relatives. All individuals with diverticular disease were excluded from the control list.

Single SNP genotyping in the replication cohort was performed at deCODE genetics with the Centaurus (Nanogen) platform[48]. The rs761545809 indel was typed using a PCR-based method using NED-labelled (yellow fluorescent dye-labelled primer, Applied Biosystems) primers. An internal size standard was added to the resulting PCR products and the fragments were separated and detected on an Applied Biosystems Model 3730 Sequencer, using in-house Allele Caller Software. Test for association in the Danish replication samples was done using logistic regression implemented in the NEMO Software[49]. The results from the replication were combined with the discovery results using a Mantel–Haenszel model[50].

**Expression analysis.** *RNA sequencing. Preparation of Poly-A cDNA sequencing libraries.* Isolated total RNA samples were assessed for quality and quantity using the Total RNA 6000 Nano Chip for the Agilent 2100 Bioanalyser. We generated cDNA libraries derived from Poly-A mRNA using Illumina's TruSeq RNA Sample Prep Kit. Briefly, using hybridization to Poly-T beads we isolated Poly-A mRNA from total RNA samples (1–4 µg input). The Poly-A mRNA was fragmented at 94 °C, and first-strand cDNA prepared using SuperScript II Reverse Transcriptase (Invitrogen) and random hexamers, followed by second-strand cDNA synthesis, end repair, addition of a single A base, adaptor ligation, AMPure bead purification and PCR amplification. The resulting cDNA was measured using the DNA 1000 Lab Chip on a Bioanalyser.

*Sequencing.* We used using Illumina's cBot and the TruSeq PE Cluster Kits v2 to cluster the samples on to flow cells. Then, we performed paired-end sequencing with either HiSeq 2000 Instruments using TruSeq v3 Flow Cells/SBS Kits or GAIIx Instruments using the TruSeq SBS Kits v5 from Illumina. Read lengths were $2 \times 125$ cycles.

*Read alignment.* We aligned the RNA sequencing reads to Homo Sapiens (Build 38) with TopHat version 2.0.12 with a set of known transcripts supplied in GTF format (RefSeq hg38; Homo sapiens, NCBI, build 38). TopHat was configured in such a way to first attempt to align reads to the provided transcriptome, following, for reads that do not map fully to the transcriptome, an attempt to map them onto the genome.

*RNA-seq quality control.* RNA libraries were excluded if the number of mapped reads were $< 10^7$ or number of mapped read pairs were $< 10^6$ or if the mapping rate of the first or second read end fell below 80% relative to the mapping of the other read end. Genotype concordance was determined by comparing imputed genotypes to those derived from RNA-seq. Samples surpassing exclusion had median 103 million mapped reads.

*RNA transcript expression.* Transcript abundance was estimated with kallisto[51] version 0.43 using Ensembl r87 transcriptome with subset to transcripts annotated as GENECODE Basic or Transcript support level 1. Transcripts with minimum five counts in each sample for at least 47% of the samples were included in the downstream analysis. Association between sequence variants and log-transformed transcripts abundances (transcripts per million) was tested on samples from the whole blood ($n = 2,947$) and adipose tissue ($n = 766$) using linear regression model with sequencing covariates listed in RNA exon expression analysis.

*RNA exon expression.* Fragments basepair aligning to exons were counted and scaled in terms of exon length and sequenced library size. Association between sequence variants and normalized expression was tested on samples from whole blood ($n = 2,246$) and adipose tissue ($n = 708$) using linear regression model with sequencing covariate terms: (1) fragment length mean, (2) exonic mapping rate, (3) number of genes detected. For blood samples, covariate terms: (4) sample preparation method and (5) read length was included. For association with adipose base libraries, the covariate terms: (4) number of alternative alignments, (5) number of mapped pairs and (6) percentage of coding bases were included.

**ROS production test.** Phagocytosis assay was performed with the Phagoburst Kit (Glycotope-Biotechnology) using concentrations of reagents and incubation times according to the manufacturer's protocol. Whole blood of 12 heterozygote carriers of the sequence variant, 14 homozygotes and 14 non-carriers as controls was stimulated with *E. coli* (particulate stimulus), phorbol 12-myristate 13-acetate (high stimulus), *N*-formyl-MetLeuPhe (low physiological stimulus) or without stimulus to serve as a negative background control. To quantify reactive oxygen metabolites, the fluorogenic substrate dihydrorhodamine 123 was used and the reaction stopped by the addition of a lysing solution that removes erythrocytes and results in partial fixation of leukocytes. The fluorescent signal was measured with a BD FACSCalibur Flow Cytometer.

**Screening for overlap with regulatory regions.** To identify which associated variants might have regulatory effects, we selected the lead variant in each locus and searched with Haploreg 4.1[52] for SNPs in LD $r^2 > 0.6$ that overlapped with predicted enhancers, DNase I hypersensitivity clusters and H3K4me1, H3K27ac and H3K9ac chromatin state assignments.

**PRS for IBD in diverticular disease.** PRS were calculated using publicly available summary statistics of IBD in Europeans from an Immunochip GWAS study on IBD[53]. Summary statistics of the European subcohort, available at https://www.ibdgenetics.org/downloads.html (downloaded on 25 October 2016), were used to assign weights to SNPs. PLINK 1.9 was used to prune SNPs in a sliding window of 500 kb, retaining the SNP that showed the strongest evidence of association with the phenotype in the training data and removed SNPs having $r^2 > 0.1$ with that SNP. We excluded the extended MHC region (chr6:25,000,000–35,000,000) from the PRS calculations due to the complex linkage disequilibrium in the region. A set of 960 whole-genome sequenced Icelanders, unrelated at six meioses served as LD reference. We calculated a polygenic score for each individual, in the target data at two different $P$ value inclusion thresholds, $P < 5 \times 10^{-8}$ and $P < 0.05$. Each PRS was then tested for association with each disease using generalized additive regression with smoothed age, sex and the first five principal components as covariates. $P$ values were adjusted for population stratification estimated by calculating association statistics from 10,000 randomly chosen SNPs with MAF $> 5\%$ and variance explained was estimated using Nagelkerke's pseudo-$R^2$ (Supplementary Table 6).

**Data availability.** The lead variants for all association signals with $P$ values less than two orders of magnitude above the relevant class-specific Bonferroni threshold are given in Supplementary Table 3a–c. Other relevant data are available from the corresponding authors on reasonable request.

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

## Acknowledgements

We thank all study participants who provided data for this study and our valued colleagues who contributed to data collection and phenotypic characterization of clinical samples, genotyping and analysis of genome sequences data. This study was funded in part by the National Institute on Drug Abuse (NIDA) (R01-DA017932).

## Author contributions

S.S., U.T., D.F.G., T.S., I.J. and K.S. designed the study, coordinated the project and interpreted the results. S.S., B.F., E.S., A.N.-C., J.B., J.A., H.S.J., E.P.-M., H.U., M.M., T.R., T.T., I.J. and T.S. coordinated and managed collection of samples and ascertainment of phenotype data. S.G. and A.S. performed experiments and analysed results. S.S., K.F.A., P.S., G.H.H., S.O., G.Th., G.M. and D.F.G. performed statistical and bioinformatic analysis. S.S., D.F.G., I.J. and K.S. drafted the manuscript. All authors contributed to the final version of manuscript.

## Additional information

**Competing interests:** S.S., K.F.A., P.S., S.G., G.H.H., S.O., A.S., T.R., T.T., G.Th., G.M., U.T., D.F.G., I.J. and K.S. are employees of deCODE Genetics/Amgen Inc. The remaining authors declares no competing financial interests.

