## [Peer Review File · Nature Communications]

Reviewer #1 (Remarks to the Author):

This is an interesting study on genetic association with diverticular disease and diverticulitis in the Icelandic population, with a replication study in Danish samples. This is a disease with little information on genetic associations, and this is an important contribution to understanding its molecular components - but it would be better placed in a specialized journal.

The selection and number of controls in the Icelandic study is a bit unclear. The abstract specifies 380,822 controls for diverticular disease and 382,013 for diverticulitis. It is not clear why the number of controls increased by this amount (it seems that some of the "diverticular" cases are now controls - which seems sub-optimal). Table 1 specifies a different number of controls, around 246-247K - a puzzling loss of controls in only few pages. The methods do not describe the choice of controls.

The study uses an innovative weighted Bonferroni adjustment where different thresholds are assigned to different variant classes. It is not clear in the Table and in the plots, which classes the highlighted SNPs belong to - and thus it is not clear why some of the variants were designed for replication. For example: rs67153654 in FAM155A with $p=8 \times 10^{-5}$ is not within "within 2 orders of magnitude from genome-wide significance threshold in" the large variant class. There is no description that puts this variant in a different class. The paper needs a clear description of the classes where the variants belong to including clarification in the Figures and Table.

Diverticulitis was not diagnosed in the replication samples - so all the diverticulitis associations have not been replicated. The authors can claim "diverticular disease replication of association", but the discussion on diverticulitis needs to be modified. In particular, the middle part of Table 1 needs to be modified; what is the meaning of the combined analysis of "Icelandic Diverticulitis + Danish diverticular disease follow up" and how is that different interpretation than "Icelandic Diverticular disease + Danish diverticular disease follow up"?

Reviewer #2 (Remarks to the Author):

SUMMARY

Sigurdsson et al. performed a GWAS using ~21.6M genetic variants identified from whole-genome sequenced or genotyped/imputed data from ~385,000 Icelanders to identify loci associated with diverticular disease and diverticulitis, a severe form of diverticular disease. Replication of 16 top sequence variants was performed in ~9000 Danish samples. Meta-analysis showed significant association of variants in the genes ARHGAP15 and COLQ with diverticular disease; and FAM155A with diverticulitis. The authors performed a nice study that is of interest to those studying diverticular disease and GI/immunity researchers more broadly, and the manuscript is generally well written. The manuscript is strengthened by eQTL and experimental results although their findings are negative. Addressing some questions and additional data would strengthen the draft.

MAJOR

1. The authors should more clearly present subject data and the rationale for study design. A table with general subject (case/control) characteristics has not been provided but is essential background information to interpret study results and its potential generalizability/comparison to future genetics studies of diverticular disease. At a minimum, please provide gender, and age. Ideally, also provide BMI, smoking history, and relevant comorbidities. This table should also show differences between those with diverticular disease vs. diverticulitis.
2. It seems as though the authors are using the diverticular disease cases without diverticulitis as controls in the diverticulitis GWAS, and pooling the diverticular disease only and diverticulitis cases as “diverticular disease” cases for the main GWAS. The rationales for these choices should be clearly stated. If the goal is to identify variants that lead to diverticulitis among those who have diverticular disease, then a nested GWAS should be performed among those with diverticular disease using as cases those with diverticulitis and controls those with diverticular disease only. Some of those with diverticular disease may one day have diverticulitis (or they may have it already and this is not captured properly in the medical record leading to some misclassification), and thus, the GWAS for diverticulitis should be performed while excluding those with diverticular disease (i.e., removing those with an intermediate phenotype). Because the Danish cohort cannot differentiate the two and because those with diverticulitis also have diverticular disease, then it is understandable why for the primary GWAS the authors used all diverticular disease subjects as cases (i.e., those with diverticular disease and diverticulitis). While the analyses listed in Table 1 partly address these issues (the FAM155A findings are very interesting), the control pool should remain constant, or an alternative nested analysis should be performed.
3. It is not clear throughout the manuscript what the sample sizes of the GWAS are. For example, the abstract lists sample size for diverticulitis cohort of “2,716 cases, 382,013 controls” (Line 45), while Table 1 lists sample sizes of 2,662 cases and 246,541 controls). The discrepancy might be due to quality control but a full description of the filtering procedure was not provided.

4. The MAFs in Table 1 should be listed by case/control status.

5. The weighted Bonferroni procedure should be explained in a few sentences despite having a reference since the procedure seems a bit arbitrary. It makes intuitive sense that “high-impact” variants are more likely to be functional than others, but exactly how to apply a weight is subjective. The authors should note that applying a full conservative Bonferroni correction to the entire set of SNPs (i.e. p threshold 2.32×10^{-9}) would still result in a significant finding at ARHGAP15 and COLQ.

6. The RNA-Seq analysis is outdated. The authors should consider a newer aligner than TopHat and an expression analysis that is transcript-based, rather than exon based. How old is the RNA-Seq data, and is it in the public domain (no GEO ID was provided)? It seems old or not sequenced very deeply if 10^7 was the threshold for sample inclusion based on mapped reads and/or some samples had mapping rate $<80\%$ to hg38. This may or may not influence their expression results.

MINOR

1. Line 65, the authors should provide a quantitative heritability measure rather than state “strong genetic component.”

2. The information of Danish cohort in Table 1 is redundant. The authors may figure out a better way to present.

3. Supplementary Fig. 2 appears ahead of Supplementary Fig. 1 in the manuscript.

4. Line 158-159: “None of the 14 missense variants found in the FAM155A gene associate with diverticular disease or diverticulitis (Supplementary Table 3)”. In Supplementary Table 3 the number of missense variants provided is 16.

Reviewers' comments:

Reviewer #1 (Remarks to the Author):

This is an interesting study on genetic association with diverticular disease and diverticulitis in the Icelandic population, with a replication study in Danish samples. This is a disease with little information on genetic associations, and this is an important contribution to understanding its molecular components - but it would be better placed in a specialized journal.

1.

The selection and number of controls in the Icelandic study is a bit unclear. The abstract specifies 380,822 controls for diverticular disease and 382,013 for diverticulitis. It is not clear why the number of controls increased by this amount (it seems that some of the "diverticular" cases are now controls - which seems sub-optimal). Table 1 specifies a different number of controls, around 246-247K - a puzzling loss of controls in only few pages. The methods do not describe the choice of controls.

Reply: We apologise that the number of controls in the abstract and tables were wrong, the same controls were used for all diverticular disease sample sets which has now been corrected.

We have recently increased our Icelandic dataset with more sequenced (15,220 vs 8,453) individuals and more chip-typed individuals (151,677 vs 150,656). We decided to use this larger sample set and updated all results in the manuscript. The new analysis resulted in a small increase in the number of used cases 5,426 vs 5,292 and a small decrease in the number of controls used 245,951 vs 246,151 (due to imputing information available) but has only minor effects on the association results. P-values in the text, Table 1 and supplementary tables have been updated accordingly. The same controls are used for all the diverticular disease sample sets.

For clarity we have replaced “diverticular disease without diverticulitis” by “uncomplicated diverticular disease” throughout the manuscript.

A more detailed information on the Icelandic cohort has been added to the methods section, **page 11 lines 6-14:**

“Phenotype data was available for 5,777 individuals with diverticular disease, including 2,923 with the primary diagnosis of diverticulitis and 2,854 with uncomplicated diverticular disease. Patients that came to the hospital primary for diverticulitis complications or if the diagnosis was coupled to a resection of the left colon or sigmoid colon were classified as diverticulitis. Genotype information was available for 94% of the diverticular disease cases or 2,764 of the 2,923 with diverticulitis and 2,662 of 2,854 uncomplicated diverticular disease. The diverticular disease patients were 60% female, with the mean age of 67.7 years (SD=4.7) and mean BMI of 27.8 (SD=5.2). Information on the patients is summarized in Supplementary Table 1.”

A more detailed description of the association analysis has been added to the methods, **page 13, line 11-18:**

“Association testing for the case-control analysis was performed using logistic regression, treating disease status as the response and genotype counts as covariates. Other available individual characteristics that correlate with disease status were also included in the model as nuisance variables. These characteristics are sex, county of birth, current age or age of death (first and second order terms included), blood sample availability for the individual and an indicator function for the overlap of the lifetime of the individual with the timespan of phenotype collection [11, 39, 40].

2.

The study uses an innovative weighted Bonferroni adjustment where different thresholds are assigned to different variant classes. It is not clear in the Table and in the plots, which classes the highlighted SNPs belong to - and thus it is not clear why some of the variants were designed for replication. For example: rs67153654 in FAM155A with $p=8 \times 10^{-5}$ is not

“within 2 orders of magnitude from genome-wide significance threshold in” the large variant class. There is no description that puts this variant in a different class. The paper needs a clear description of the classes where the variants belong to including clarification in the Figures and Table.

Reply. We agree with the reviewer and we have added clarification of the variant classes to the methods, page 12 lines 20 to page 13 line 11:

“We used the Variant Effect Predictor (VEP)[37] to predict the maximal consequence of each sequence variant on all neighboring RefSeq genes[38]. There is a substantial variation in the enrichment of phenotype-associating sequence variants based on their annotations [12]. On the basis of these enrichments, it is possible to group sequence variants into categories, in order of decreasing impact on biological function. We used the enrichment of variant classes to correct the threshold for genome-wide significance with a weighted Bonferroni adjustment [12]. With 32,463,443 sequence variants tested, the weights given in Sveinbjornsson et al. were rescaled to control the family-wise error rate (Supplementary Table 2). This yielded significance thresholds of $P < 2.6 \times 10^{-7}$ for high-impact variants (N=8,474, including stop gained, frameshift, splice acceptor or donor), $P < 5.1 \times 10^{-8}$ for moderate-impact variants (N=149,983, including missense, splice-region variants and in-frame INDELs), $P < 4.6 \times 10^{-9}$ for low-impact variants (N=2,283,889 (including synonymous variants 3' and 5' UTR variants), $P < 2.3 \times 10^{-9}$ for other variants overlapping DNase hypersensitivity sites (N=3,913,058) and $P < 7.9 \times 10^{-10}$ for other non-DNase hypersensitivity site variants, intergenic and deep intronic (N=26,108,039) [12] (Supplementary Table 2).”

rs67153654 has a $P = 1.7 \times 10^{-7}$ for association with diverticulitis (second part of table 1), which for that phenotype is within the 2 orders of magnitude from the threshold of 2.3×10^{-9} for the variant class it belongs to. We refer to the variant classes in supplementary table 3a-c in the text, page 4, lines 14-17:

We chose sixteen variants for follow up in a Danish diverticular disease sample set that were within two orders of magnitude from genome-wide significance threshold in a variant class for **either** diverticular disease or diverticulitis in Iceland (Table 1, Supplementary Table 3a-c).

and qualify the class of the significant variants in the results, **page 4, line 20 to page 5 line 1:**

"With these datasets we identified three loci that **are** of genome wide-significance in the combined analysis of the Icelandic and Danish samples; **intronic** variants at the *ARHGAP15* and *COLQ* loci associate significantly with diverticular disease and at *FAM155A* locus with diverticulitis (**$P < 2.3 \times 10^{-9}$** , the threshold for intronic variants **within a DNase hypersensitivity site [12]**) (Table 1, **Supplementary Table 3**)"

The reference has also been corrected (**page 4, line 14**, ref.12 Sveinbjornsson et al).

The variant classes are given for all the variants in Supplementary Table 3.

We have clarified the description of the classes the variants described fall into **in the legends of Figure 1 and Table 1:**

"The three variants rs4662344, rs7609897 and rs67153654 are annotated as intronic-variant within a DNase hypersensitivity site, giving the class-specific Bonferroni threshold for genome wide significance as $p < 2.3 \times 10^{-9}$ [12]"

3. Diverticulitis was not diagnosed in the replication samples - so all the diverticulitis associations have not been replicated. The authors can claim "diverticular disease replication of association", but the discussion on diverticulitis needs to be modified. In particular, the middle part of Table 1 needs to be modified; what is the meaning of the combined analysis of "Icelandic Diverticulitis + Danish diverticular disease follow up" and how is that different interpretation than "Icelandic Diverticular disease + Danish diverticular disease follow up"?

Reply: Diverticulitis is the severe form of diverticular disease and therefore we analyzed separately variants that show stronger association with diverticulitis than diverticular disease as a whole in the Icelandic cohort. The FAM155A variant rs67153654 shows a suggestive association with diverticular disease ($P = 8.7 \times 10^{-5}$, OR=0.89) but larger effect and stronger association with diverticulitis ($P = 2.3 \times 10^{-7}$, OR=0.80), but no association with uncomplicated diverticular disease ($P = 0.78$, OR=0.99). This indicates that this variant may contribute to progression and severity of the disease rather than to risk of getting the disease. This is however difficult to validate since diverticulitis diagnosis in the Danish cohort is not available. Diverticulitis is reported to be 5-20% of diverticular disease (Sheth 2008) but is 51.9% in the Icelandic diverticular disease sample. Some of the Danish diverticular disease subjects are bound to have diverticulitis, but we do not know whether the fraction is similar

to that in Iceland. Accordingly, using the Danish diverticular disease sample for follow up of the diverticulitis signal is not optimal, but justifiable, since it dilutes the diverticulitis subset with samples from those who have uncomplicated diverticular disease, and thus decreases our chance of replicating the signal. This “dilution” is consistent with the effect of rs67153654 in Denmark (OR=0.84), in between that of diverticular disease overall and diverticulitis in Iceland. Despite these limitations the combined Danish diverticular disease sample, which includes an unknown proportion of diverticulitis, the combined Icelandic diverticulitis sample and Danish diverticular disease sample shows genome wide significant association with rs67153654 ($P=3.0 \times 10^{-11}$, OR=0.82). See also response to reviewer 2, #2 below.

This has been clarified in the introduction (**page 4, lines 17-20**):

“We do not have information on diverticulitis in the Danish cohort, but chose to follow up the Icelandic diverticulitis findings based on the assumption that the Danish cohort includes diverticulitis, although the proportion is unknown.”

in the discussion (**page 7, lines 16-20**):

“Although diagnosis of diverticulitis is not available for the Danish samples we genotyped the *FAM155A* variant in the Danish diverticular disease samples. We replicated the association with rs67153654-A (Danish diverticular disease: OR=0.84, $P=2.2 \times 10^{-5}$) despite the lack of information on the proportion of diverticulitis among the Danish diverticular disease samples.”

in the methods (**page 12, line 4**):

information on the proportion of diverticulitis was not available

and in the legend of Table 1:

*Note that the Danish diverticular disease results are the same in all three parts of the table, since diverticulitis diagnosis was not available.

Reviewer #2 (Remarks to the Author):

SUMMARY

Sigurdsson et al. performed a GWAS using ~21.6M genetic variants identified from whole-genome sequenced or genotyped/imputed data from ~385,000 Icelanders to identify loci

associated with diverticular disease and diverticulitis, a severe form of diverticular disease. Replication of 16 top sequence variants was performed in ~9000 Danish samples. Meta-analysis showed significant association of variants in the genes ARHGAP15 and COLQ with diverticular disease; and FAM155A with diverticulitis. The authors performed a nice study that is of interest to those studying diverticular disease and GI/immunity researchers more broadly, and the manuscript is generally well written. The manuscript is strengthened by eQTL and experimental results although their findings are negative. Addressing some questions and additional data would strengthen the draft.

MAJOR

1. The authors should more clearly present subject data and the rationale for study design. A table with general subject (case/control) characteristics has not been provided but is essential background information to interpret study results and its potential generalizability/comparison to future genetics studies of diverticular disease. At a minimum, please provide gender, and age. Ideally, also provide BMI, smoking history, and relevant comorbidities. This table should also show differences between those with diverticular disease vs. diverticulitis.

Reply: We agree with the reviewer and have added a Supplementary Table 1 with age, sex, BMI and smoking as suggested, although the information is not available for all subjects. We have also improved the information on the subjects in the methods section (**page 11, lines 10-14**):

“Genotype information was available for 94% of the diverticular disease cases or 2,764 of the 2,923 with diverticulitis and 2,662 of 2,854 uncomplicated diverticular disease. The diverticular disease patients were 60% female, with the mean age of 67.7 years (SD=4.7) and mean BMI of 27.8 (SD=5.2). Information on the patients is summarized in Supplementary Table 1.”

2. It seems as though the authors are using the diverticular disease cases without diverticulitis as controls in the diverticulitis GWAS, and pooling the diverticular disease only

and diverticulitis cases as “diverticular disease” cases for the main GWAS. The rationales for these choices should be clearly stated.

If the goal is to identify variants that lead to diverticulitis among those who have diverticular disease, then a nested GWAS should be performed among those with diverticular disease using as cases those with diverticulitis and controls those with diverticular disease only.

Some of those with diverticular disease may one day have diverticulitis (or they may have it already and this is not captured properly in the medical record leading to some misclassification), and thus, the GWAS for diverticulitis should be performed while excluding those with diverticular disease (i.e., removing those with an intermediate phenotype)

Reply:

All the diagnosis come from the two main hospitals in Iceland and have been reviewed by a single specialist. We acknowledge that those who do not have a diverticulitis diagnosis might develop it later or possibly have had it without the diagnosis being captured. We still believe that it is helpful to look at the disease associating variants among those who have had have diverticulitis diagnosis and for comparison those who do not. We also point out that over half of the diverticular disease cohort has diverticulitis. We argue that the stronger association of rs67153654 with diverticulitis and larger effect than with diverticular disease overall and the complete lack of association with uncomplicated diverticular disease, strongly suggests that the effect of the variant is on progression or severity rather than the risk of developing the disease per se.

The rational for testing diverticular disease phenotypes overall is now better explained in the introduction, **page 4, lines 17-20** (as explained in response to reviewer’s 1 comment 3):

“We do not have information on diverticulitis in the Danish cohort, but chose to follow up the Icelandic diverticulitis findings based on the assumption that the Danish cohort includes diverticulitis, although the proportion is unknown.”

and in the discussion (**page 7, lines 16-20**):

”Although diagnosis of diverticulitis is not available for the Danish samples we genotyped the *FAM155A* variant in the Danish diverticular disease samples. We replicated the association with rs67153654-A (Danish diverticular disease: OR=0.84,

$P=2.2 \times 10^{-5}$) despite the lack of information on the proportion of diverticulitis among the Danish diverticular disease samples. “

We emphasize that the same controls are used for diverticular disease, diverticulitis and uncomplicated diverticular disease (see response to reviewer 1, comment 1), which is now clearly stated in the text (**page 4, line 10-12**):

“performed two GWAS to search for sequence variants that affect the risk of diverticular disease (5,426 cases) and diverticulitis (2,764 cases) using the same 245,951 controls (Supplementary Table 1)”.

in the legend of Figure 1

(cases $n=2,764$ cases, controls $n=245,951$) (excluding diverticular disease)

in the legend of Table 1:

*Note that the Danish diverticular disease results are the same in all three parts of the table, since diverticulitis diagnosis was not available.

and in the methods **page 14, line 1-8**

A total of 5,426 individual with the diverticular disease were included in the association analysis; 3,368 of these were genotyped using various Illumina chips and imputed using long-range phased haplotypes and the remaining 1,958 were imputed on the basis of genotypes of 1st and 2nd degree relatives [11]. The same population controls, 245,951 individuals recruited through different deCODE project, were used for association analysis of the three diverticular disease phenotypes: 124,228 genotyped and 121,723 imputed on the basis of genotypes of 1st and 2nd degree relatives. All individuals with diverticular disease were excluded from the control list.

A nested GWAS is another approach, but has limited power compared to testing the two subphenotypes against the same very large control sample. We have however, tested the association of the 3 reported variants on diverticulitis vs. uncomplicated diverticular disease and shown there is a significant difference in the allele frequency for the FAM155A variant, **page 8 line 10-12**:

“We tested the association of diverticulitis versus uncomplicated diverticular disease for these three variants (Supplementary Table 4). The FAM155A variant is significantly less frequent in diverticulitis (OR=0.84, 95% CI (0.74,0.94), $P=3.8 \times 10^{-3}$)”

3. Because the Danish cohort cannot differentiate the two and because those with diverticulitis also have diverticular disease, then it is understandable why for the primary GWAS the authors used all diverticular disease subjects as cases (i.e., those with diverticular disease and diverticulitis). While the analyses listed in Table 1 partly address these issues (the FAM155A findings are very interesting), the control pool should remain constant, or an alternative nested analysis should be performed.

Reply: As explained above (#2) the same controls were used for diverticular disease, diverticulitis, and uncomplicated diverticular disease (see also response to reviewer 1, #1), We have updated the numbers of samples used in the text and tables.

4. It is not clear throughout the manuscript what the sample sizes of the GWAS are. For example, the abstract lists sample size for diverticulitis cohort of “2,716 cases, 382,013 controls” (Line 45), while Table 1 lists sample sizes of 2,662 cases and 246,541 controls). The discrepancy might be due to quality control but a full description of the filtering procedure was not provided.

Reply: All the numbers have been corrected in the abstract, text, Table 1, and supplementary tables (see response to reviewer 1, #1),

4. The MAFs in Table 1 should be listed by case/control status.

Reply: MAF among cases and controls are only available for the directly genotyped samples not for the familialy imputed part of the dataset. We have thus included the MAF information based on the directly genotyped samples to the Table 1 and Supplementary Table 3, with an explanatory note in the legends:

“Minor allele frequency is calculated on the chip-typed samples, excluding familialy imputed genotypes”

5. The weighted Bonferroni procedure should be explained in a few sentences despite

having a reference since the procedure seems a bit arbitrary. It makes intuitive sense that “high-impact” variants are more likely to be functional than others, but exactly how to apply a weight is subjective. The authors should note that applying a full conservative Bonferroni correction to the entire set of SNPs (i.e. p threshold 2.32×10^{-9}) would still result in a significant finding at ARHGAP15 and COLQ.

Reply: The weighted Bonferroni procedure is explained in the methods page 12, line 20 page 13 line 3:

“We used the Variant Effect Predictor (VEP)[37] to predict the maximal consequence of each sequence variant on all neighboring RefSeq genes[38]. There is a substantial variation in the enrichment of phenotype-associating sequence variants based on their annotations [12]. On the basis of these enrichments, it is possible to group sequence variants into categories, in order of decreasing impact on biological function. We used the enrichment of variant classes to correct the threshold for genome-wide significance with a weighted Bonferroni adjustment [12]. “

Supplementary Table 2 describes the classes of variants and P-value thresholds for each class. We have clarified the description of the classes in the legends of Figure 1 and Table 1:

“The three variants rs4662344, rs7609897 and rs67153654 are annotated as intronic-variant within a DNase hypersensitivity site, giving the class-specific Bonferroni threshold for genome wide significance as $P < 2.3 \times 10^{-9}$ [12]”

6. The RNA-Seq analysis is outdated. The authors should consider a newer aligner than TopHat and an expression analysis that is transcript-based, rather than exon based. How old is the RNA-Seq data, and is it in the public domain (no GEO ID was provided)? It seems old or not sequenced very deeply if 10^7 was the threshold for sample inclusion based on mapped reads and/or some samples had mapping rate $< 80\%$ to hg38. This may or may not influence their expression results.

Reply: We agree with reviewer that Tophat aligner is becoming outdated. Therefore we have added a new transcript based analysis with larger set of RNA-seq samples, see methods (page 15, lines 19 to page 16 line 2) and boxplots in Supplementary Figures 1a, 3a, 4a):.

"RNA transcript expression. Transcript abundance was estimated with kallisto [46] version 0.43 using Ensembl r87 transcriptome with subset to transcripts annotated as GENECODE Basic or Transcript support level 1. Transcripts with minimum five counts in each sample for at least 47% of the samples were included in the downstream analysis. Association between sequence variants and log transformed transcripts abundances (transcripts per million) was tested on samples from whole blood (n=2947) and adipose tissue (n=766) using linear regression model with sequencing covariates listed in RNA exon expression analysis."

Supplementary Figure 2a-c have been redone. The results are now shown in three supplementary figures, one for each variant and gene 1a-b (rs4662344/ARHGAP15), 3a-b (rs7609897 /COLQ), and 4a-b (rs67153654 /FAM155A), showing the effects of the variants on expression of both a) multiple transcripts and b) individual exons.

The following statement on transcript expression analysis has been added to the main text added,

and Page 5 line 21:

" (Supplementary Fig. 1, shown for each transcript and each exon)."

The RNA samples are of high quality. We have improved the section of RNA-seq quality control in methods (**page 15, lines 14-18**):

"RNA-seq quality control. RNA libraries were excluded if the number of mapped reads were less than 10^7 or number of mapped read pairs were less than 10^6 or if the mapping rate of the first or second read-end fell below 80% relative to the mapping of the other read-end. Genotype concordance was determined by comparing imputed genotypes to those derived from RNA-seq. Samples surpassing exclusion had median 103 million mapped reads".

The data is recent and has not been made publicly available.

Other changes:

Paragraph in the methods section, page 3 on RNA expression with microarrays deleted as it had been replaced with RNA sequencing results in the manuscript.

MINOR

1. Line 65, the authors should provide a quantitative heritability measure rather than state “strong genetic component.”

Reply: The paragraph has been revised to include heritability estimates from the references, **page 4, lines 3-5:**

“Relative risk of siblings of diverticular disease cases is 2.9[8] and estimated the heritability in twin-studies is 40%-50%[8, 9]. This indicates that there is a strong genetic component to the risk.”

2. The information of Danish cohort in Table 1 is redundant. The authors may figure out a better way to present.

Reply: We agree that this is difficult to present, and have added the following explanation in the legend to Table 1:

*Note that the Danish diverticular disease results are the same in all three parts of the table, since diverticular disease diagnosis was not available.

3. Supplementary Fig. 2 appears ahead of Supplementary Fig. 1 in the manuscript.

Reply: We thank the reviewer for noting, and we have renumbered the supplementary figures, as they appear in the text.

4. Line 158-159: “None of the 14 missense variants found in the FAM155A gene associate with diverticular disease or diverticulitis (Supplementary Table 3)”. In Supplementary Table 3 the number of missense variants provided is 16.

Reply: The correct number is 16 and has been corrected in text, **page 8 line 14**

Note that we have made minor changes in the text for clarification, these are highlighted. We have also removed the presentation of the variant in HECW1 to the Supplementary note, since it does not reach genome wide significance.

Note that Gudmar Thorleifsson who performed much of the additional statistical analysis has been added as a coauthor.

Reviewer #1

{{EDITOR: This referee did not return formal comments to the authors as s/he found the revised paper to be satisfactory and endorses publication}}

Reviewer #2 (Remarks to the Author):

The paper is much improved. Now that the case/control numbers throughout are consistent and details on the method used have been added, the analysis is much easier to understand.